# BNT162b2 COVID-19 Vaccines in Children, Adolescents and Young Adults with Cancer—A 1-Year Follow-Up

**DOI:** 10.3390/vaccines11050989

**Published:** 2023-05-16

**Authors:** Caroline Donze, Victoria Min, Laetitia Ninove, Xavier de Lamballerie, Gabriel Revon Rivière, Arnauld Verschuur, Paul Saultier, Nicolas André

**Affiliations:** 1Department of Pediatric Hematology, Immunology and Oncology, APHM, La Timone Children’s Hospital, 13000 Marseille, France; caroline.donze@ap-hm.fr (C.D.); victoria.min@ap-hm.fr (V.M.); gabriel.revon-riviere@ap-hm.fr (G.R.R.); arnauld.verschuur@ap-hm.fr (A.V.); paul.saultier@ap-hm.fr (P.S.); 2Unité des Virus Émergents, UVE Aix Marseille University, IRD 190, INSERM 1207, APHM, 13005 Marseille, France; laetitia.ninove@ap-hm.fr (L.N.); xavier.de-lamballerie@univ-amu.fr (X.d.L.); 3INSERM, INRAe, C2VN, Aix Marseille University, 13005 Marseille, France; 4CNRS, INSERM, Institut Paoli-Calmettes, CRCM, Aix Marseille University, 13009 Marseille, France

**Keywords:** COVID vaccine, immunity, cancer, pediatrics

## Abstract

(1) Background: Children and young adults with cancer are poorly represented in COVID-19 vaccination studies, and long-term protection conferred by vaccination is not known. (2) Objectives: 1. To determine the adverse effects associated with BNT162B2 vaccination in children and young adults with cancer. 2. To assess its efficacy in stimulating immunological response and in preventing severe COVID-19 disease. (3) Methods: Retrospective single-center study evaluating patients aged 8 to 22 years, with cancer, who underwent vaccination from January 2021 to June 2022. ELISA serologies and serum neutralization were collected monthly from the first injection. Serologies below 26 were considered negative, while those above 264 BAU/mL were considered positive and indicative of protection. Antibodies titers were considered positive above 20. Data on adverse events and infections were collected. (4) Results: 38 patients were included (M/F = 1.7, median age 16 years), of whom 63% had a localized tumor and 76% were undergoing treatment at the time of the first vaccination. Two or three vaccine injections were administered in 90% of patients. Adverse events were mainly systemic and not severe, except for seven grade 3 toxicities. Four cancer-related deaths were reported. Median serology was negative the month following the first vaccination and became protective during the third month. At 3 and 12 months, median serology was 1778 and 6437 BAU/mL, respectively. Serum neutralization was positive in 97% of the patients. COVID-19 infection occurred despite vaccination in 18%; all were mild forms. (5) Conclusions: In children and young adults with cancer, vaccination was well tolerated and conferred effective serum neutralization. COVID-19 infections were mild, and vaccine seroconversion persisted after 12 months in most patients. The value of additional vaccination should be further established.

## 1. Introduction

Severe acute respiratory syndrome coronavirus 2 (SARS-CoV-2)-related coronavirus disease 2019 (COVID-19) particularly affects patients with certain chronic pathologies. For instance, adults with cancer develop more severe COVID-19 forms with a higher rate of mortality challenging clinicians for the management of these patients [1]. In children, the observed incidence seems much lower than in adults, very likely because children are asymptomatic or with mild symptoms [2,3]. However, children with cancer are more likely to develop a severe form of the disease [2,4,5,6]. In addition, even mild or moderate infections may impact the management of their disease, for instance by delaying or modifying treatment.

As China has changed its “zero-COVID” strategy, a new wave of COVID-19 is rapidly spreading, further highlighting the importance of COVID-19 vaccine [7]. Indeed, vaccine remains the major preventive treatment for COVID-19. Its efficacy has been demonstrated in immune-compromised adult patients and in patients with cancer who present less acute respiratory complications after vaccination [8,9]. Regarding the higher risk for severe COVID-19 and the available knowledge in adults with cancer, the benefits of vaccination were shown to outweigh the potential harms despite the need for more vaccine injections to acquire a lasting immune response. Although data are now available or under study regarding vaccination in children and young adults, few studies are reported in pediatric oncology and, to the best of our knowledge, no long-term data have been published yet concerning children with cancer [10,11,12].

Here, we report the long-term results of a single-center retrospective cohort of children, adolescents and young adults, undergoing or shortly after completing their cancer treatment, who received BNT162b2 vaccine. The objectives of the study were to determine the adverse effects associated with BNT162b2 vaccine and to assess the clinical efficacy of the SARS-CoV-2 vaccination in preventing severe COVID-19 and the natural evolution of COVID-19 serologies in children and young adults with cancer.

## 2. Materials and Methods

Population selection: All children over 5 years of age, adolescents and young adults, with a solid tumor or non-Hodgkin’s lymphoma treated in the Department of Pediatric Oncology in La Timone Hospital, AP-HM in Marseille and eligible for vaccination against COVID-19, were offered a vaccine injection of BNT162b2, from 20 January 2022 for patients over 12 years of age and from 1 January 2022 for children over 5 years of age, until 30 June 2022. Patients with these criteria who accepted vaccination were enrolled.

Vaccination recommendations: Patients were to receive three doses of BNT162b2 vaccine (30 µg per dose) according to “Haute Autorité de Santé” recommendations (21 days between the first 2 doses, then from 3 months after the primary vaccination) [13]. BNT162b2 vaccine is a lipid nanoparticle-formulated, nucleoside-modified RNA vaccine that encodes a prefusion-stabilized, membrane-anchored SARS-CoV-2 full-length spike protein [14].

Data collection: Demographic data (gender, date of birth, living or deceased status) and medical data (pathology, localized or metastatic status, number of lines of treatment, type of treatment during vaccination, date of end of treatment, dates of vaccinations), medical data regarding adverse vaccine events (according to the CTCAE classification), severity and symptoms of COVID-19 infection if any (according to the WHO classification of COVID-19 severity described in Appendix A) were retrieved from the electronic medical records of patients.

Serological analysis: Biological and immunological data were collected from serum samples taken before each vaccine injection and monthly until data freeze. Serum samples were tested by an ELISA serological test of anti-SARS-CoV-2 IgG antibodies directed against the S1 domain of the virus spike protein with a positivity threshold of 264 BAU/mL. Anti-SARS-CoV-2 IgG antibodies directed against the S1 domain of the virus Spike protein were assessed using the QuantiVac ELISA kit from Euroimmun^®^ (Lubeck, Germany). Thus, serologic immunity and maintained immunity were defined by the acquisition and maintenance of a serology above 264 BAU/mL, as measured by an ELISA serological test. The threshold of 264 BAU/mL was accepted as the cut-off point for seropositivity based on the results of the study by Feng et al. [15]. Neutralizing antibodies against the SARS-CoV-2 viral strain were assessed with a microneutralization test [16]. The test used clinical strains of SARS-CoV-2 (100 TCID50/well), TMPRSS2-expressing VeroE6 cells and relied on cytopathic effect (CPE) detection at 5 days post-infection. It was a VNT100 (100% of wells lysed in duplicate format). The test was automated in a NSB3 laboratory for all dilution and dispensing steps and for CPE reading. Dilutions tested were 20, 40, 80, 160, 320, 640 and 1280. The range was extended if a titer of 1280 was observed in the first instance. First, we studied the serology and serum neutralization of all patients; then we looked at the results of the subgroup of patients with non-Hodgkin’s lymphoma.

Statistical analysis: Descriptive analysis was performed using Excel (Microsoft Office Excel Spreadsheet.xlsx). Quantitative variables were expressed as proportions (%) or as median with interquartile range (Q1 and Q3). The probability of maintaining immunity against SARS-CoV-2 was estimated using the Kaplan-Meier method (GraphPad Prism 9). Alluvial diagram was generated using the sankeymatic online tool (https://sankeymatic.com/build/) (assessed 16 May 2023).

Ethics: All data have been generated as part of the routine care at Assistance Publique-Hôpitaux de Marseille (AP-HM), and this study results from routine clinical management. The study was approved by the AP-HM, and access to the patients’ biological and registry data issued from the hospital information system was approved by the data protection committee of APHM (identifier PADS21-136). All patients signed informed consent.

## 3. Results

### 3.1. Global Population

Overall, 265 patients, older than 12 years and eligible for vaccination between 1 January 2021 and 30 June 2022 or aged 5 to 12 years and eligible for vaccination between 1 January 2022 and 30 June 2022, were followed in the pediatric oncology department. Thirty-eight of them (14%) were vaccinated.

A total of 38 patients received at least 1 vaccine injection: 17 (45%) received three injections; 17 (45%) received two injections; and four (11%) received a single injection. Four patients died from the progression of their disease: Two of them received two vaccine doses, and the two others received one dose. The sex ratio (M/F) was 1.7. The median age at the first vaccination was 16.0 years [15.0; 18.0]. The cancer was localized in 24 (63%) cases. Patients had received one line of treatment in 22 cases (58%), two lines in nine cases (24%) and more than three lines in seven cases (18%). The oncological treatment consisted in systemic therapy (chemotherapy or immunotherapy) in the majority of the patients (*n* = 37; 97%) and localized by surgery, radiotherapy or radiofrequency for 29 patients (76%). The general characteristics of the patients are detailed in Table 1.

### 3.2. Adverse Events

Adverse events reported after each vaccine injection are shown in Table 2. Adverse events were described in 26 patients. Most of them were low-grade adverse events. Seven severe adverse events (fatigue) were reported. The main adverse events were local pain (dose #1: *n* = 19; 54%; dose #2: *n* = 13; 41%, dose #3: *n* = 7; 23%) and fatigue (dose #1: *n* = 7; 20%; dose #2: *n* = 10; 31%, dose #3: *n* = 8; 51%). One case of non-extensive vascular purpura without fever was reported in a woman of 20 years-old within 5 days of vaccination. This purpura remained uncomplicated and resolved without any further investigation or specific treatment.

### 3.3. COVID-19 Infections

Six COVID-19 infections were reported after the first vaccination: Four cases within 6 months after vaccination and two cases beyond the 6-month post-vaccination period. All reported COVID-19 diagnosed in the patients included in the study were mild or moderate forms. Patients were asymptomatic in 29% of cases (*n* = 2). The symptoms described were rhinorrhea (*n* = 3; 43%), coughing (*n* = 3; 43%), asthenia (*n* = 2; 29%), fever (*n* = 1, 14%), headache (*n* = 2; 29%), ageusia and anosmia (*n* = 1; 14%).

### 3.4. Immunity

The anti-S1 antibodies level increased over time after the first vaccine injection and became positive beyond the third month, as described in Figure 1A–C. Median serologies for each monthly interval are listed in Appendix A.

Indeed, as shown in Figure 1A, the median serology at 1 month of primary vaccination was positive but below the protection threshold now established as 173 [26; 498] BAU/mL. At 3 and 6 months, it remained positive above the protection threshold, respectively, at 1778.3 [1379; 3099] BAU/mL and 895.4 [367; 3219] BAU/mL. Surprisingly, at 9 months, the median serology was positive but under the protection threshold as 216 [162; 2088] BAU/mL. For eight patients for whom we had collected serologies beyond the 12th month from the first vaccine, these serologies remained positive and protective against COVID-19 infection, with a median of 5878 [696; 7137] BAU/mL.

The serologies of 15 patients who acquired post-vaccination immunity during the first 3 months after vaccination, with an anti-S1 antibody level above 264 BAU/mL, were analyzed. The probability of maintaining immunity against SARS-CoV-2 decreased over time, reaching the level of 50% at 259 days post-vaccination, as shown in Figure 1B.

Figure 1C shows, after linear regression analysis, that the anti-S1 antibody level increases with time after vaccination, exceeding 200 BAU/mL after 200 days. Each point represents a serology’s result. The anti-S1 antibody in the subgroup of patients with non-Hodgkin’s lymphoma are highlighted in gray. These serologies are lower than in patients with other solid tumors, without statistical comparison by subgroup.

The serological variations over time are shown in Figure 2. This alluvial diagram represents the evolution over time of the serologies of patients initially not immunized against COVID-19. The connections between the panels allow the seropositive or seronegative status of each patient to be followed at each month where the data is available. The COVID panel corresponds to patients who became COVID positive during the follow-up and whose vaccine serostatus can therefore no longer be interpreted after the viral infection. This COVID panel was created to avoid the false conclusion that a patient became seropositive after the vaccine when post-infectious immunity was acquired. Thus, among 26 patients with an initial negative serology, six had a COVID-19 infection; nine acquired a satisfactory immunity persisting more than 10 months after vaccination; and only one remained seronegative at 10 months after the first vaccine dose. Figure 2 emphasizes that the vaccine seroconversion remained positive between the 4th and 10th month.

Interestingly, the data showed three outlier patients who did not appropriately respond to vaccination: one patient whose serology was negative at month 0 became positive after month 10; one patient whose serology was positive at month 3 then negative at month 10; one patient whose serology was positive at month 2 then became negative at month 4 then became positive again after month 7. The first two patients had completed their treatment and had been monitored for more than 3 years, while the third one was undergoing maintenance treatment by Temodal, 3 months after the end of radiotherapy, for a glioma.

Serum neutralization was positive (titer ≥ 20) at least once in 97% (*n* = 31) of patients for whom this data was available (*n* = 32). The titer was positive in 83% (10/12) at first month after vaccination, 100% (6/6) at 3 months, 100% (5/5) at 6 months, 100% (3/3) at 9 months and 100% (3/3) at 12 months.

## 4. Discussion

The BNT162b2 vaccine is the major preventive treatment for the SARS-CoV-2, of which children and young adults with cancer are more likely to develop a severe form. However, this subgroup is poorly represented in COVID-19 vaccination studies, and only very limited data is available [11,12,17].

Our cohort is currently the largest study analyzing the clinical and immunological impact of COVID-19 vaccination conducted in children, adolescents and young adults with cancer. It also has the longest follow-up with 1-year follow-up. We report that patients developed a protective immunity against COVID-19 infection, at 3 months. Serum neutralization was positive at least once in almost all cases (97%), reflecting in vitro efficacy of the vaccination in our sample of patients. These findings are consistent with those of Lehrnbecheret et al., who reported their experience in 21 pediatric cancer patients who received three doses of SARS-CoV-2 mRNA vaccine (BioNTech/Pfizer) [12]. They showed that vaccines elicited both humoral and cellular immunity in most patients. Interestingly, patients remained protected after 10 months. Data from our study shows that the probability of losing immunity acquired through vaccination increases over time, exceeding 50% at 259 days post-vaccination. These results raise the question of the interest of additional vaccine injection.

It had already been demonstrated that, after two doses of vaccine in healthy children aged 5 to 11 years, the COVID-19 neutralizing titers showed a satisfactory immunogenicity [18]. In adult patients with cancer who were vaccinated during anticancer treatment, seropositivity after BNT162b2 vaccination was achieved after two doses, but the antibody response was inferior compared to healthy adult controls [19]. In a 2021 study in adults with solid tumor, the pattern of immunogenicity and efficacy of the BNT162b2 vaccine six months post-vaccination looks like the pattern of the general adult population [8]. Our data here confirms that children with cancer develop stronger anti-COVID immunity. On the other hand, post-vaccination serological response in patients with lymphoma may be different from that observed in patients with other solid tumors. In our subgroup of non-Hodgkin’s lymphoma patients, the response to the vaccine appears to be less good based on serologies. However, the limited number of patients in our population does not allow for statistical comparison by subgroup. In the study led by Della Pia et al., adult lymphoma patients appeared to be able to develop a humoral response to vaccines [20].

Six out of 38 developed COVID-19 despite vaccination. We can speculate that this number is likely underestimated as some cases of asymptomatic COVID-19 forms may not have been diagnosed. All reported forms were mild or moderate according to the WHO classification (WHO/2019-nCoV/clinical/2021.2). Mukkada et al., have reported through an international cohort gathering data from 131 institutions in 45 countries that, among 1500 pediatric cancer patients, before COVID immunization, 259 (20%) of 1301 patients had a severe or critical infection, and 50 (4%) of 1319 died with the cause attributed to COVID-19 infection [6]. Hence, although children with cancer are usually asymptomatic or have mild forms, the vaccine may prevent severe and critical forms that account for 19% [21]. Altogether, these data suggest that SARS-CoV-2 mRNA vaccine decreases COVID-19 severity in children with cancer. This is consistent with data from healthy children in whom, out of 48 patients, seven mild or moderate COVID-19 infections were reported [8,18]. In adults with solid tumor, out of 154 patients, only one patient had a documented COVID-19 infection after the second vaccine dose (a severe illness that required hospitalization).

Interestingly, in two patients (5%), the post-vaccination serologies remained negative despite repeated vaccine injections but then became clearly positive after a COVID-19 infection. Therefore, these two patients did not acquire post-vaccinal immunity but highly post-infectious immunity. The specific reasons for this discrepancy regarding the ability to induce immunity remains unknown. However, several studies report that natural immunity acquired after COVID-19 infection remains superior to post-vaccination immunity [22,23,24]. For example, the study by Gazit et al., emphasizes that naturally acquired immunity is stronger than after vaccination, especially against the Delta variant [22]. Moreover, Lewis et al., have nevertheless reported the efficacy of vaccination even in patients previously infected with COVID-19 [23]. Postvaccination immunity declined 6 months after the vaccine injection, whereas infection-acquired immunity declined only after 1 year in unvaccinated participants, but remained greater than 90% in those subsequently vaccinated, according to the results of Hall et al. [24].

Most of the adverse events we found after vaccination were low grade and had been already previously reported. The systemic events (fatigue, pain, fever, gastrointestinal symptoms) are similar to those observed among healthy adults and children [18,25]. Unlike adults or children with history of anaphylaxis, no allergic reactions were reported [26]. Emergency protocols have been established for systemic allergic reactions that may occur in children with allergies, asthma or immunodeficiency [27]. Only one unexpected effect was reported: a purpura that occurred five days after the vaccine. The platelet count was unknown at the time of occurrence of the event. Purpura resolved spontaneously. While we were not able to determine whether the purpura was related to the vaccine administration, several cases of acquired thrombotic thrombocytopenic purpura have already been reported in adults within several days after receiving the NBT162b2 vaccine [27]. A small series by Maayan et al., described four patients with low ADAMST-13 activity treated with plasma exchanges. Treatment was successful in three patients [28]. This post-vaccination vascular phenomenon is reminiscent of the vaccine-induced immune thrombotic thrombocytopenia (VITT) associated with ChAdOx1 nCoV-19 vaccine [29]. The immunological mechanism involved seems to be the production of antibodies that bind to PF4 and form immune complexes that induce platelet activation, whose most serious manifestation is cerebral venous sinus thrombosis [29,30]. However, the VITT syndrome has never been reported after BNT162b2 administration [31].

In our cohort, only 14% of patients and families accepted the vaccination. This percentage is much lower than those found in other countries for the global pediatric population. For example, in November 2021, the US CDC recommended Pfizer’s COVID-19 messenger RNA vaccine for children between 5 and 11 years old, i.e., 28 million children. The US surveys show that a very high percentage ranging from 42% to 66% of parents of these children had refused this vaccination [32], confirming the need to further communicate about both the safety and efficacy of the vaccine in reducing morbidity and mortality of COVID-19 infection. To address this low percentage of people vaccinated, communication strategies have been implemented. While it is well demonstrated that vaccine acceptance is a major factor in controlling the pandemic, the messages currently being developed have not proven effective [33,34,35,36]. Ongoing international prospective studies will provide relevant data in the frailest populations to better inform the worldwide COVID-19 vaccination campaign.

There are several limitations to our study. First, our results should be confirmed in a multi-center setting with a higher number of patients. Second, both the small number of patients, the difficulty in collecting all serologies monthly and the high percentage of vaccine refusals induces selection bias that does not allow for definitive conclusions. Indeed, serum samples were meant to be collected at each month, but in practice, we got the analysis approximatively quarterly after month 3. The effect of age and number of vaccine doses should be investigated in larger studies. Lastly, the correlation between positive serology and effective immunity or conversely between negative serology and weakened immunity remains debated. To date, there is no reliable correlation between clinical efficacy and post-vaccine immune response, either in cancer patients or in the general population, as highlighted by the study by Fendler et al. [37].

## 5. Conclusions

At least one dose of BNT162b2 vaccine confers protection against severe COVID-19 with satisfactory immunogenic effect and possible clinical benefit at 1 year follow-up in children and young adults under treatment for a solid tumor. The majority of patients received two doses of vaccine, suggesting that the immunogenic characteristics described for at least one dose could be similar in patients who received two vaccine doses; however, this hypothesis must be confirmed by further studies. Vaccine seroconversion appears to persist longer than in adults although the design of our study does not allow for a significative comparison of child and adult populations. The potential need for additional boosters should be further established. Furthermore, the results of this study could help to reassure patients and family who have concerns about the BNT162b2 vaccine and thus reduce the high rate of vaccine refusal.

## Figures and Tables

**Figure 1 vaccines-11-00989-f001:**
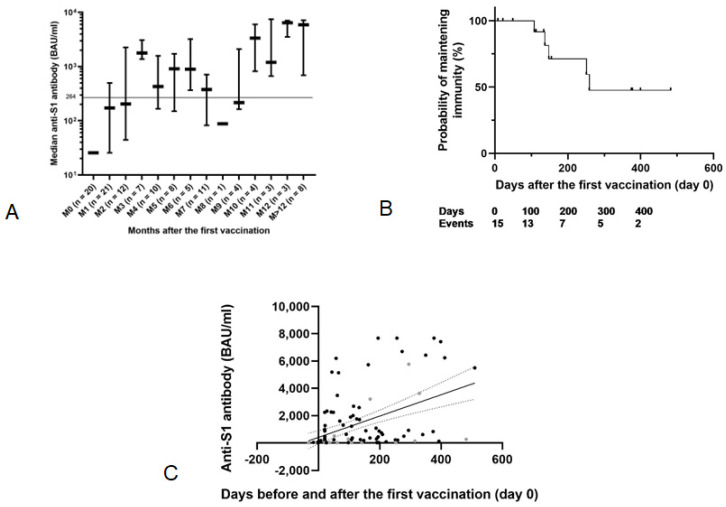
(**A**) Evolution of the anti-S1 antibody levels over time measured by Elisa test. Antibody levels are expressed as median levels with IQR. Patients with documented COVID-19 infection have been removed (11 patients). (**B**) Probability of maintaining immunity over time after the first vaccine injection using the Kaplan-Meier method. The 15 patients who acquired post-vaccination immunity during the first three months after vaccination are included in this analysis. (**C**) Kinetics of humoral response after the first vaccination with Spearman-type linear regression (R^2^ = 0.22; *p* < 0.0001).

**Figure 2 vaccines-11-00989-f002:**
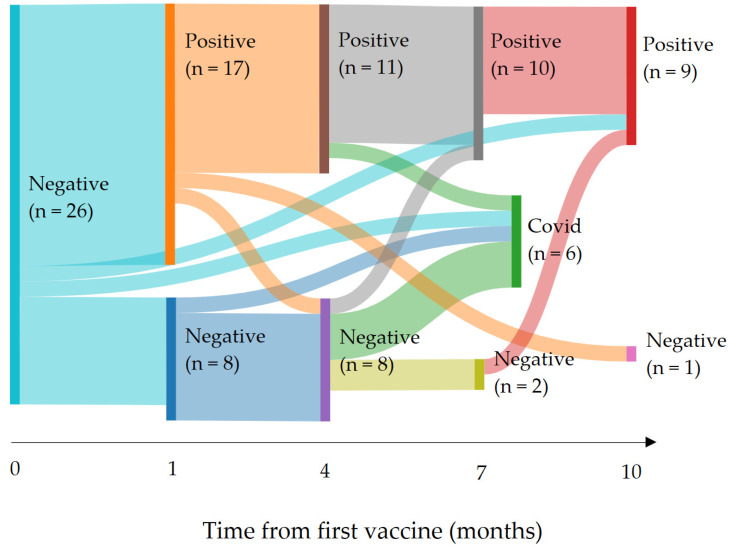
Serological and COVID-19 status evolution COVID: COVID-19 infection (serological data are removed after COVID-19 infection). The width of each bar is proportional to the number of patients involved. The COVID panel corresponds to the patients having contracted a COVID infection with a positive PCR test during the follow-up period. The follow-up time after the first vaccine injection represented on the abscissa is expressed in months. Each Positive panel is a group of patients with serology above the threshold of 264 BAU/mL. Conversely, each Negative panel corresponds to those with serology below this threshold.

**Table 1 vaccines-11-00989-t001:** Patients’ characteristics.

		General Population N = 38	
		N or Median [Q1; Q3]	Percentage
**Sex**			
	Female	14	37%
	Male	24	63%
**Number of administered vaccine doses**		N	%
	1	4	N/A
	2	17	N/A
	3	17	N/A
**Age at the first vaccination**		16.0 [15.0; 18.0]	
**Death**			
	Alive	34	90%
	Deceased	4	11%
**Type of cancer**			
	Brain tumor	9	24%
	Lymphoma	7	18%
	Bone sarcoma	6	16%
	Soft tissue sarcoma	10	26%
	Other types	6	16%
**Metastatic status**			
	Localized	24	63%
	Metastatic	14	37%
**Types of treatment**			
**Systemic**		37	97%
	Conventional chemotherapy	36	95%
	High dose chemotherapy	5	14%
	Immunotherapy	11	29%
**Local**		29	76%
	Surgery	23	61%
	Radiotherapy	18	47%
	Radiofrequency	1	3%
**Number of lines**			
	1	22	N/A
	2	9	N/A
	>3	7	N/A
**Treatment phase at the time of vaccination**			
	Induction	1	N/A
	Adjuvant	6	N/A
	Maintenance	16	N/A
	Palliative chemotherapy	6	N/A
	Surveillance	9	N/A

**Table 2 vaccines-11-00989-t002:** Vaccination-related adverse events.

		Vaccine 1 N = 35	Vaccine 2 N = 32	Vaccine 3 N = 16
None		13 (37%)	14 (44%)	5 (31%)
Local pain	Grade 1	19 (54%)	13 (41%)	7 (23%)
Diffuse pain				
	Grade 1	2 (6%)	2 (6%)	0
	Grade 2	0	0	1 (6%)
Fatigue				
	Grade 1	5 (14%)	5 (16%)	3 (19%)
	Grade 2	1 (3%)	2 (6%)	2 (13%)
	Grade 3	1 (3%)	3 (9%)	3 (19%)
Fever	Grade 1	0	1 (3%)	1 (6%)
Vomiting	Grade 2	0	1 (3%)	0
Purpura		0	0	1 (6%)
Dizziness		0	0	1 (6%)

N = 35 for the first vaccine dose (38 vaccinated, 3 missing data); N = 32 for the second (34 vaccinated, 2 missing data); N = 16 for the third (17 vaccinated, 1 missing data).

## Data Availability

Data will be provided upon reasonable request to authors.

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
