# Peer review of "BNT162b2 COVID-19 Vaccines in Children, Adolescents and Young Adults with Cancer—A 1-Year Follow-Up"

_vaccines, 2023, doi:10.3390/vaccines11050989_

Round 1
Reviewer 1 Report
In the present scenario, to understand the aftermath effect of post-Covid- 19 Vaccine administration specifically in cancer pathology is very significant. The current manuscript entitled “BNT162b2 COVID-19 Vaccines in Children, Adolescents, and Young Adults with Cancer - A 1 Year Follow-Up” by Donze et al illustrated the follow-up studies of BNT162b2 Covid-19 vaccination in children and young adults with cancer, was well endured and conferred effective serum neutralization. In addition, the results of this study could support encouraging patients and families who have concerns about the BNT162b2 vaccine and thus diminish the high rate of vaccine refusal. The results are interesting, and accurate and are comprehensively discussed in the present manuscript. However, some of the following concerns need to be addressed by the author.
1. At first, what is the basis, children had to receive the three doses of BNT162b2 COVID-19 Vaccines in the present study. This should be addressed with a proper citation because this vaccine booster dose was used in children over five years of age, adolescents, and young adults, a solid tumor or non-Hodgkin’s lymphoma.
2. Another limitation of the study as the author stated in the discussion population size is low when it comes to other vaccinations. These studies will be more conclusive if the population size is more. so further studies and data need to be included
3. In Figure 1. Legend statistical information needs to be incorporated, because Figure. 1 has multiple analysis, for the reader's point of view descriptive statistical information needs to be included in the figure legend.
4. In Sentence 270, proper citation needs to be incorporated to support the statement.
5. In Sentence 43, the following (recent article) citation might be helpful to justify the statement. (Regression of Lung Cancer in Mice by Intranasal Administration of SARS-CoV-2 Spike S1) PMID: 36428739.
Overall English is good, but the authors need to improve minor English editing in the final format of the current manuscript.
Author Response
In the present scenario, to understand the aftermath effect of post-Covid- 19 Vaccine administration specifically in cancer pathology is very significant. The current manuscript entitled “BNT162b2 COVID-19 Vaccines in Children, Adolescents, and Young Adults with Cancer - A 1 Year Follow-Up” by Donze et al illustrated the follow-up studies of BNT162b2 Covid-19 vaccination in children and young adults with cancer, was well endured and conferred effective serum neutralization. In addition, the results of this study could support encouraging patients and families who have concerns about the BNT162b2 vaccine and thus diminish the high rate of vaccine refusal. The results are interesting, and accurate and are comprehensively discussed in the present manuscript. However, some of the following concerns need to be addressed by the author.
- At first, what is the basis, children had to receive the three doses of BNT162b2 COVID-19 Vaccines in the present study. This should be addressed with a proper citation because this vaccine booster dose was used in children over five years of age, adolescents, and young adults, a solid tumor or non-Hodgkin’s lymphoma.
The citation for national vaccine recommendations in this population is line 73 https://www.has-sante.fr/jcms/p_3178533/fr/vaccination-dans-le-cadre-de-la-COVID-19. We have re-phrased the citation correctly in the manuscript (13).
- Another limitation of the study as the author stated in the discussion population size is low when it comes to other vaccinations. These studies will be more conclusive if the population size is more. so further studies and data need to be included.
There is a paucity of data regarding COVID‐19 vaccination in children with cancer, we currently haven’t found any further study/reference to add in bibliography. We have highlighted this point in the discussion. Furthermore our cohort is the largest, which confirms the lack of data and the importance of this work.
- In Figure 1. Legend statistical information needs to be incorporated, because Figure. 1 has multiple analysis, for the reader's point of view descriptive statistical information needs to be included in the figure legend.
We agree with Reviewer 1 and have added descriptive statistical information in the legend.
- In Sentence 270, proper citation needs to be incorporated to support the statement.
This is the same citation as in the following sentence line 272 [27] Maayian et al.. For clarity, we have added it on line 270.
- In Sentence 43, the following (recent article) citation might be helpful to justify the statement. (Regression of Lung Cancer in Mice by Intranasal Administration of SARS-CoV-2 Spike S1) PMID: 36428739.
We are unable to link line 43 to this article. Moreover, lung cancer is very rare in pediatrics and extrapolation of in vivo results in mice is very speculative.

Reviewer 2 Report
1. "ELISA serologies and serum neutralization were collected monthly from the first injection" vs "Figure 1(A)" vs "Table S1" vs "Table S2".
- Were the sera collected monthly or quarterly? Please indicate the actual procedure to ensure data validity and amend the related sentences and presentations accordingly. If the sera could not be collected monthly as anticipated, it can be reported as a limitation and can be discussed under results.
2. Materials and Methods- please improve formatting, in which each paragraph should have a subtopic to represent the work. "Population" and "Data" can be combined as e.g. "Sample recruitment", while specific serological tests should fall in a separate subtopic, e.g. "Serological analysis". Overall, this section requires major re-arrangements and corrections.
3. Table 1: change % to Percentage.
4. Table 1: Percentages for "Number of administered vaccine doses", "Number of lines" and "Treatment phase at the time of vaccination" might not be suitable to be presented as percentage. The values can be a standalone and should be discussed without percentages. Indicate N/A (Not applicable) in the Table.
5. Improve caption for Figure 2, i.e. include the meaning of COVID panel, time/duration, and what does it mean by negative/positive. It was a good attempt to summarise data into an "interactive" alluvial diagram, however, the caption needs to be crystal clear to ensure the it is a standalone and not relying on the text for clarification.
All the best
Author Response
- "ELISA serologies and serum neutralization were collected monthly from the first injection" vs "Figure 1(A)" vs "Table S1" vs "Table S2".
- Were the sera collected monthly or quarterly? Please indicate the actual procedure to ensure data validity and amend the related sentences and presentations accordingly. If the sera could not be collected monthly as anticipated, it can be reported as a limitation and can be discussed under results.
Serum samples were meant to be collected at each month during the first 3 months and more spaced out later on. Therefore in practice, they were not collected at each month for each patient during all the follow-up period. As suggested, we added and discussed this limitation line 294.
- Materials and Methods- please improve formatting, in which each paragraph should have a subtopic to represent the work. "Population" and "Data" can be combined as e.g. "Sample recruitment", while specific serological tests should fall in a separate subtopic, e.g. "Serological analysis". Overall, this section requires major re-arrangements and corrections.
We thanks the reviewer for the suggestion. We have re-arranged and corrected this section accordingly.
- Table 1: change % to Percentage.
We have done the modifications accordingly.
- Table 1: Percentages for "Number of administered vaccine doses", "Number of lines" and "Treatment phase at the time of vaccination" might not be suitable to be presented as percentage. The values can be a standalone and should be discussed without percentages. Indicate N/A (Not applicable) in the Table.
We agree with Reviewer 2 and have indicated N/A in the Table.
- Improve caption for Figure 2, i.e. include the meaning of COVID panel, time/duration, and what does it mean by negative/positive. It was a good attempt to summarise data into an "interactive" alluvial diagram, however, the caption needs to be crystal clear to ensure the it is a standalone and not relying on the text for clarification.
The Covid panel corresponds to the patients having contracted a Covid infection with a positive PCR test during the follow-up period. The follow-up time after the first vaccine injection represented on the abscissa is expressed in months. Each Positive panel is a group of patients with serology above the threshold of 264 BAU/ml. Conversely, each Negative panel corresponds to those with serology below this threshold.
We have added all these explanations in the caption of Figure 2.

Reviewer 3 Report
Donze et al. conducted a study on the effect of COVID-19 vaccines in children and young adults with cancer. The authors observed that serum neutralization was positive in almost all cases, indicating that vaccination was well tolerated in young patients with cancer and resulted in effective serum neutralization. While the subject matter and conclusions of this manuscript are noteworthy, as they emphasize the effectiveness of COVID-19 vaccination and the need to address the high rate of vaccine refusal in young patients with cancer, there is room for improvement in the presentation of this manuscript. Please see my comments below:
(1) Line 19-20: The sentence in the abstract, "Serologies were considered negative below 26, and positive above 264 BAU/ml," reads confusingly. Perhaps the authors could rephrase it to say something like "Serologies below 26 were considered negative, while those above 264 BAU/ml were considered positive and indicative of protection."
(2) Line 117: The word "tree" appears to be a typo.
(3) Line 166: Please reconsider the word choice "exceeding" since the number falls below 50%.
(4) Figure 1C: The correlation between the regression line and the real data points seems to be weak. Further commentary is needed on this observation.
(5) Line 181-182: The authors mentioned that "only two remained seronegative at 10 months after the first vaccine dose," while Figure 2 indicates that only one patient remained seronegative at the 10-month mark. Please provide clarification on the discrepancy between the main text and the figure.
(6) Line 232: The authors stated that "seven out of 38 developed COVID-19 despite vaccination," whereas Figure 2 shows six patients developed COVID-19. Please explain the difference between the main text and the figure.
The English usage is acceptable. However, as I mentioned in my previous comments, the authors should thoroughly proofread the manuscript and improve the clarity of its presentation.
Author Response
Donze et al. conducted a study on the effect of COVID-19 vaccines in children and young adults with cancer. The authors observed that serum neutralization was positive in almost all cases, indicating that vaccination was well tolerated in young patients with cancer and resulted in effective serum neutralization. While the subject matter and conclusions of this manuscript are noteworthy, as they emphasize the effectiveness of COVID-19 vaccination and the need to address the high rate of vaccine refusal in young patients with cancer, there is room for improvement in the presentation of this manuscript. Please see my comments below:
(1) Line 19-20: The sentence in the abstract, "Serologies were considered negative below 26, and positive above 264 BAU/ml," reads confusingly. Perhaps the authors could rephrase it to say something like "Serologies below 26 were considered negative, while those above 264 BAU/ml were considered positive and indicative of protection."
We agree with Reviewer 3 and have changed the sentence accordingly.
(2) Line 117: The word "tree" appears to be a typo.
We have corrected typo in the manuscript.
(3) Line 166: Please reconsider the word choice "exceeding" since the number falls below 50%.
We have changed “exceeding” into “reaching the level of”.
(4) Figure 1C: The correlation between the regression line and the real data points seems to be weak. Further commentary is needed on this observation.
The regression line is significant with R2 = 0.22 and p < 0.0001. We have added these numbers in the Figure 1C legend.
(5) Line 181-182: The authors mentioned that "only two remained seronegative at 10 months after the first vaccine dose," while Figure 2 indicates that only one patient remained seronegative at the 10-month mark. Please provide clarification on the discrepancy between the main text and the figure.
We thank Reviewer 3 for pointing out this typo. Only one patient remained seronegative at the 10 months. We have corrected this in the manuscript.
(6) Line 232: The authors stated that "seven out of 38 developed COVID-19 despite vaccination," whereas Figure 2 shows six patients developed COVID-19. Please explain the difference between the main text and the figure.
Again, we thank for his careful review Reviewer 3 for pointing out this typo. Six patients developed COVID-19 but seven severe adverse events were reported. We have corrected this mistake in the manuscript.
